# In Situ Growth of Nickel–Cobalt Metal Organic Frameworks Guided by a Nickel–Molybdenum Layered Double Hydroxide with Two-Dimensional Nanosheets Forming Flower-Like Struc-Tures for High-Performance Supercapacitors

**DOI:** 10.3390/nano13030581

**Published:** 2023-01-31

**Authors:** Cheng Cheng, Yongjin Zou, Fen Xu, Cuili Xiang, Lixian Sun

**Affiliations:** Guangxi Key Laboratory of Information Materials, Guilin University of Electronic Technology, Guilin 541004, China

**Keywords:** layered double hydroxide, metal organic framework, nanoparticles, hybrid supercapacitor

## Abstract

Metal organic frameworks (MOFs) are a kind of porous coordination polymer supported by organic ligands with metal ions as connection points. They have a controlled structure and porosity and a significant specific surface area, and can be used as functional linkers or sacrificial templates. However, long diffusion pathways, low conductivity, low cycling stability, and the presence of few exposed active sites limit the direct application of MOFs in energy storage applications. The targeted design of MOFs has the potential to overcome these limitations. This study proposes a facile method to grow and immobilize MOFs on layered double hydroxides through an in situ design. The proposed method imparts not only enhanced conductivity and cycling stability, but also provides additional active sites with excellent specific capacitance properties due to the interconnectivity of MOF nanoparticles and layered double hydroxide (LDH) nanosheets. Due to this favorable heterojunction hook, the NiMo-LDH@NiCo-MOF composite exhibits a large specific capacitance of 1536 F·g^−1^ at 1 A·g^−1^. In addition, the assembled NiMo-LDH@NiCo-MOF//AC asymmetric supercapacitor can achieve a high-energy density value of 60.2 Wh·kg^−1^ at a power density of 797 W·kg^−1^, indicating promising applications.

## 1. Introduction

The increasing depletion of non-renewable resources and serious damage to the environment indicate that the research and development of new, excellent electrochemical energy storage systems with high-capacity, environmental protection, and low-cost factors are increasingly important [1]. Supercapacitors are considered to be ideal energy storage devices as they have the characteristics of high power density, rapid charge–discharge process, long-term stability, and long-period performance [2]. Because of the different charge storage mechanisms, supercapacitors are divided into electric double-layer capacitors (EDLCs) and pseudo-capacitors. Electric double-layer capacitors, also known as non-Faraday capacitors, achieve energy storage through the accumulation of free electrons at the interface between the material and electrolyte. In contrast, pseudo-capacitors store charges through a rapid reversible redox reaction and intercalation process on the electrode surface. Therefore, it has a larger specific capacitance than non-Faraday capacitors. However, due to the insertion/extraction of redox reactive ions on the surface of the active material, the structure produces defects during continuous expansion and compression processes until collapse, so the cyclic stability of the pseudo-capacitor is slightly lower than that of the EDLC. At the same time, due to the limitations of the dual-layer or surface Faraday storage mechanism and voltage window, their low-energy density limits the wide application of supercapacitors. According to the formula (E = 1/2CV^2^), the operating voltage and specific capacitance determine the energy density of the supercapacitor [3]. In addition, the electrode material determines the electrochemical performance of the supercapacitor. Therefore, the design and optimization of electrode material structures are the key to obtain high-energy density. Generally, two methods are used to increase the energy density of electrode materials. One method involves combining the active material with a highly conductive material, while the other is to optimize the structure or composition of the active material.

Metal organic frameworks (MOFs) as porous organic materials have attracted considerable interest in the field of electrochemical energy storage in recent years [4,5] This is due to their tunable microstructure, porous properties, high specific surface area, and ease of synthesis [6,7]. Because of their excellent electrochemical properties, MOFs have great application prospects as pseudo-capacitors. Various MOFs can be manufactured by adding Ni, Co, Mn, and other metal ions to achieve an optimal electrochemical performance [8]. Among them, Co-MOFs and Ni-MOFs have been extensively studied for their excellent electrochemical properties [9]. The results show that Ni-MOFs have energy storage capacities, while Co-MOFs have good reversibility properties. Therefore, with the synergistic effect of Ni and Co metal ions, Ni and Co bimetallic organic skeletons have better electrochemical properties than monometallic organic skeletons. Conventional MOFs have long diffusion pathways and expose fewer active sites. Therefore, to overcome these drawbacks, MOFs can be prepared through a method that confers hollow structures, fast transport pathways, and abundant redox reaction sites [10]. However, freestanding MOFs are prone to restacking during preparation, unstable, and tend to agglomerate. To address these limitations, we designed and synthesized composite materials of MOFs to improve the electrochemical properties of single materials and strengthen the stability of the structure. Recently, Shi et al. synthesized interconnected an NiCo-LDH@MOF nanosheet structure with excellent electrochemical properties (approximately 723 C·g^−1^ at 1 A·g^−1^) [11]. These experiments indicated that the addition of appropriate second-phase materials can not only adjust the morphology of MOFs, enhance the stability of the structure, but also provide additional capacitance.

Layered double hydroxides (LDHs) are two-dimensional functional materials with specific layered structures and different chemical compositions [12]. They are a promising material for use in electrodes owing to their relatively high redox activity, high stability, and low toxicity [13]. A nickel–molybdenum layered double hydroxide (LDH) contains the transition metals Ni and Mo, can provide more active sites, and has greater electrochemical activity and higher specific capacitance than its single hydroxide. In particular, the structural design of LDHs improves the electrochemical performance by adjusting the nano/microstructure morphology, including ultrathin nanostructures and specific three-dimensional and nano-network structures; it can improve the electron conductivity and ion mobility in the redox process [14,15]. The interconnected layered nanosheets can construct the open nanosphere structure, provide more nanochannels, and allow the diffused and permeated OH^−^ ions move freely, which is conducive to improving the conductivity. Ultrathin nanostructures and layered structures coordinate with each other to prevent the accumulation of nanosheets internally, and to enhance more surface electrochemical activity externally. Therefore, LDHs can be used as a dispersed and oriented carrier material [16,17].

In this work, we devise a simple and effective method to inhibit the restacking of hollow-structured MOFs by immobilizing them on flower-like NiMo-LDH nanosheets with a layered structure by a two-step hydrothermal solution. Curling NiMo-LDH nanosheets can provide A·greater surface area and expose more active sites. There is a certain space between the curved nanosheets, which can avoid the serious agglomeration of nanosheets and reduce the transmission path of electrolyte acceleration ion diffusion. LDHs can be used as dispersive and oriented carrier materials, which can fix and disperse MOF nanoparticles through the metal–carrier interaction and modify the structural properties of MOF nanoparticles [18]. With the coordination of the support structure and surface composition, NiMo-LDH@NiCo-MOF as an electrode can provide a high-performance capacitance of 1536 F·g^−1^ at 1 A·g^−1^. An asymmetric capacitor assembled as a positive electrode with NiMo-LDH@NiCo-MOF can achieve a high-energy density value of 60.2 Wh·kg^−1^ at a power density of 797 W·kg^−1^.

## 2. Experimental Section

### 2.1. Synthesis of NiMo-LDH

In this study, 2 mmol Ni(NO_3_)_2_·6H_2_O (98%) and Na_2_MoO_4_·2H_2_O (99.90%) were added to 50 mL of deionized water, and 15 mmol urea (99%) was poured in after stirring well. After stirring the mixture for half an hour, the reaction mixture was transferred to a Teflon-lined stainless-steel autoclave and held at 120 °C for 4 h. After the autoclave was cooled to room temperature, the collected products were washed three times with ethanol and deionized water, and dried overnight under vacuum at 80 °C to obtain green powder.

### 2.2. Synthesis of NiMo-LDH@NiCo-MOF composites

A total of 50 mg of NiMo-LDH powder was weighed and dispersed ultrasonically in a mixed solution of ethanol (10 mL), N,N-Dimethylformamide (10 mL), and deionized water (10 mL). Then, 0.75 mmol of trimesic acid (98%), 2 g of polyvinylpyrrolidone (k29-32), 0.25 mmol of Co(NO_3_)_2_ 6H_2_O (98.5%), and 1.25 mmol of Ni(NO_3_)_2_ 6H_2_O (98%) were added into the solution and stirred for half an hour to disperse the resulting solution. The solution was transferred to an autoclave and heated to 150 °C for 10 h. Once at room temperature, the sample was repeatedly rinsed with ethanol and finally dried overnight at 80 °C. The sample denoted as NiMo-LDH@NiCo-MOF. NiCo-MOF was prepared under the same reaction conditions. NiCo-MOF was obtained by the same procedure as mentioned above. The sample preparation flow chart is shown in Figure 1.

## 3. Results and Discussion

The morphologies of NiMo-LDH, NiCo-MOF, and NiMo-LDH@NiCo-MOF samples were studied by SEM, as shown in Figure 2a,b, which show that NiMo-LDH is a standard three-dimensional flower-like structure consisting of two-dimensional curved nanosheets. These interconnected layered nanosheets were assembled into a 3D nanoflower structure with an open spatial structure [19]. This structure provides large accessible surfaces with multiple active sites for redox reactions. Figure 2c,d show that NiCo-MOF has a spherical, hollow structure. The formed hollow structure not only provided more open space and abundant surface reaction sites to reduce the diffusion length of ions and electrons, but also buffers the structural changes caused by the electrochemical process, so as to show an excellent electrochemical performance [10]. Figure 2e,f are the SEM images of NiMo-LDH@NiCo-MOF, showing that NiCo-MOF is immobilized on flower-like NiMo-LDH with a layered structure. The use of NiCo-LDH as the support material provides the conditions for the orientation and stable growth of the MOF, generating highly dispersed and homogeneous active sites that can perform redox reactions rapidly with a higher specific capacitance. The morphology of NiMo-LDH@NiCo-MOF following the cycle is shown in Appendix A. Although the structure of NiMo-LDH@NiCo-MOF following circulation has presents little caking and collapse, the overall structural integrity is maintained.

The phase composition of the crystal was measured by XRD. Figure 3a shows the spectrum of NiMo-LDH@NiCo-MOF. For NiMo-LDH, the diffraction peaks at 34.5° and 59.5° correspond to (012) and (110) crystal planes, respectively (ICC card number: 01-082-8040) [15]. In the XRD pattern of NiMo-LDH@NiCo-MOF, it was observed that the diffraction peaks of NiCo-MOF appearing at 2θ = 9.3°, 11.8°, 18.3°, and 23.8° correspond to the (100), (010), (200), and (020) crystal planes of NiCo-MOF, respectively, which is consistent with the NiCo-MOF card (CCDC 638866) indexed [8]. The characteristic peaks of NiCo-MOF are roughly the same as those of NiMo-LDH@NiCo-MOF, showing that NiCo-MOF nucleates and grows in situ on NiMo-LDH and does not change the structure of the crystal. This proves that NiMo-LDH has excellent stability and provides support conditions for the in situ nucleation and growth of NiCo-MOF.

The chemical states of each element in NiMo-LDH@NiCo-MOF were studied by measuring the XPS spectra. Appendix A shows the high-resolution XPS spectra of (a) Ni 2p and (b) Mo 3d for NiMo-LDH. Figure 4a shows the total spectrum of the sample used to analyze the element composition. The spectra indicated the presence of Ni, Co, O, C, and Mo elements. Among them, the peak intensity of Ni 2p was higher than that of Co 2p, indicating that the Ni content was higher and the Co content was lower. In the high-resolution spectrum, both the Ni 2p and Co 2p spectra contained two spin double peaks and two vibrational satellite peaks (Sat). The molar ratio of Ni/Co was calculated as 5:1, and the coexistence of Ni and Co was beneficial to improve the electrochemical performance. As shown in Figure 4b, for the high-resolution XPS spectrum of Ni 2p, the binding energies corresponding to the characteristic peaks of Ni 2p_1/2_ and Ni 2p_3/2_ are 873.6 and 856.1 eV, respectively, and two satellite peaks are observed at 879.9 and 861.7 eV [20]. 

For the Co 2p spectrum (Figure 4c), the binding energies corresponding to the high peaks of Co 2p_3/2_ and Co 2p_1/2_ were 781.3 and 797.3 eV, respectively, which are consistent with the characteristic peaks of NiCo-MOF [10]. The vibrational satellite peaks corresponding to the binding energies at 786.1 and 803.1 eV indicate the presence of Co^2+^. Figure 4d shows that the spectrum of Mo 3d consists of two main peaks. The spin orbits of 232.02 and 235.17 eV are Mo 3d_5/2_ and Mo 3d_3/2_, respectively [15]. The two peaks of O 1s in Figure 4e at 531.2 and 533.1 eV correspond to metal–oxygen bonds in metal oxides and oxygen in hydroxyl ions, respectively. In addition, three distinct peaks of the C 1s spectrum are located at 284.7, 285.8, and 288.4 eV [11].

More information about the morphology and structure of NiMo-LDH@NiCo-MOF composites was obtained by TEM and HRTEM. Figure 5a,b shows TEM images at different magnifications. It can be observed from the TEM images at different magnifications (Figure 5a,b) that tiny NiCo-MOF particles are scattered over the 3D flower-like surface of NiMo-LDH. The spherical structure of the NiMo-LDH nanosheets was further revealed, and the spaces between the nanosheets can store electrolytes to shorten the diffusion distance. By scanning the sharp contrast between light and dark in the TEM image, the black spots shown are NiCo-MOF loaded on the NiMo-LDH nanosheets. From the HRTEM images (Figure 4d), it can be observed that the crystal-plane spacing at 0.28 nm perfectly matches the simulated (220) crystal plane of Co-MOF [21].

The element distribution of NiMo-LDH@NiCo-MOF was analyzed by energy dispersive X-ray spectroscopy (EDS). Figure 6a–f shows that the product is composed of Co, Ni, Mo, C, O, and other elements, and each element is evenly distributed. It is clear that Ni, Co, Mo, C, and O are evenly distributed across the surface of the flower. This is consistent with their SEM images.

In order to study the electrochemical interface reaction, electrochemical impedance spectroscopy (EIS) tests were performed in the wide band range (0.01–100 kHz) [22]. Figure 7 shows the Nyquist curves of the three samples, and the frequencies in the impedance spectrum can be divided into high and low. The horizontal intercept represents the equivalent series resistance (Rs), giving the internal electrode resistance of the system and the resistance at the electrolyte/electrode interface. The charge transfer resistance (Rp) corresponds to the semicircular radius of the high frequency region and is generated by the ion transfer between the electrolyte and the electrode. The EIS diagram depicts a semicircle at high frequencies, corresponding to the charge transfer. In the low-frequency region, a slanted line of Warburg impedance is depicted, corresponding to ion mobility [23,24]. With the increase in the slope, the diffusion of the electrode material is improved. It can be observed from the figure that the gradient of the NiMo-LDH@NiCo-MOF composite material is steeper than that of NiMo-LDH and NiCo-MOF, and the electrolyte ion diffusion is better [25]. This is because the unique nanoflower structure has a better contact area and provides more ion diffusion paths, which facilitates electrolyte diffusion and ion transport. The combination of nanoflowers and nanoparticles provides convenient nanochannels. The increase in nanochannels leads to more paths for ion diffusion, and thus less electrochemical impedance.

Compared to the single component, the suitable second-phase composite has a larger specific surface area. Nitrogen (N2) adsorption and desorption data were measured at 77 K to assess the specific surface area and porosity. As shown in Figure 8a, NiMo-LDH, NiCo-MOF, and NiMo-LDH@NiCo-MOF show H4 hysteretic rings, which proves the existence of a fractured pore structure. As can be observed from the figure, the specific surface areas of NiMo-LDH, NiMo-LDH@NiCo-MOF, and NiCo-MOF were 193.2, 49.0, and 15.4 m^2^/g, respectively, as measured by Brunauer–Emmett–Teller (BET). Among them, NiMo-LDH@NiCo-MOF can increase the larger specific surface area due to the unique nanoflower structure in close contact with the nanoparticles. The obtained size distribution (Figure 8b) shows that the pore volume peaks are in the mesoporous (2–10 nm) range, indicating that the pore structure of the sample contains a high number of mesopores [26]. When the electrode is in contact with the solution, the high number of mesopores shortens the adsorption distance between ions and facilitates the transport of electrolyte ions, allowing for high-charge storage and multiplication performance [27]. The abovementioned results indicate the successful synthesis of stable NiMo-LDH@NiCo-MOF composites. In addition, a strong interaction can be detected between NiMo-LDH and NiCo-MOF, which is conducive to structural stability and electron transfer through interface effects.

In order to investigate the pseudo-capacitance performance of NiMo-LDH@NiCo-MOF electroactive materials, CV and GCD were tested in 3M KOH aqueous electrolytes. Figure 9a shows the CV curves from a low to high sweep speed [28]. Under the condition of a low scanning rate, the symmetric curve shows that there is a reversible redox reaction at the electrode/electrolyte interface, indicating that the electrode has A·good pseudo-capacitance performance [29]. As the scanning rate of the extended region increases, the curve gradually deviates from the normal state, and the redox peaks shifts to both ends of the potential window due to the decrease in the interfacial resistance, indicating rapid electron transport and ion permeation process [30,31]. Figure 9b shows the change in potential over time in the charge–discharge curves of NiMo-LDH@NiCo-MOF, and the operating voltage can reach 0.5 V. The GCD curve shows two distinct charge/discharge platforms, proving that it is a Faraday process, which also supports the CV curve analysis [32,33]. When the current density is 1 A·g^−1^, the NiMo-LDH@NiCo-MOF electrode exhibits the highest capacitance and longest discharge time. The capacitance values gradually decrease with the increasing scan rate. The reason for the decline of specific capacitance values at a high current density is that the redox reaction cannot fully react inside the active material and the electrolyte ions cannot fully contact the active site during the rapid charge–discharge process, resulting in an insufficient redox reaction. When the current density increases from 1 to 10 A·g^−1^, it is calculated that the specific capacitance of the NiMo-LDH@NiCo-MOF electrode slowly decreases from 1536 to 1060 F·g^−1^ (Figure 9c). These results show excellent capacity retention. The NiMo-LDH@NiCo-MOF electrode exhibits a higher capacity retention of 69% at 10 A·g^−1^ with respect to the initial specific capacitance, which indicates that the NiMo-LDH@NiCo-MOF electrode has a significant multiplicative performance. This is because the flower-like, spherical structure of the NiMo-LDH@NiCo-MOF composite provides a large surface area, which not only facilitates the distribution of active sites, but also increases the contact area between the active electrode and electrolyte, further improving the energy storage performance of the supercapacitor at high current densities. It can be observed from Figure 9d that the potential of the charge/discharge curves exhibit typical pseudo-capacitance behavior over time. The calculated specific capacitances for NiMo-LDH, NiCo-MOF, and NiMo-LDH@NiCo-MOF at 1 A·g^−1^ are 388, 732, and 1536 F·g^−1^, respectively. Among them, NiMo-LDH@NiCo-MOF has the best specific capacitance and the longest discharge time among the three. Figure 9e shows the closed, shaded region indicating the non-diffusion control process. Because of the effect of polarization, a part of the shaded region is outside the original CV curve. As shown in Figure 9f, the contributions of the pseudo-capacitance and diffusion control are calculated as a proportion of the total capacitance based on the closed, shaded region. This suggests that the electrochemical reaction of NiMo-LDH@NiCo-MOF is determined by both capacitance and diffusion behavior. In addition, with the increase in the scanning rate, the contribution of the diffusion control process decreases significantly, and the redox reaction gradually dominates the charge storage process, indicating that the pseudo-capacitive charge storage is dominant in the specific capacity. 

In order to study the practical application of the NiMo-LDH@NiCo-MOF active electrode, the asymmetric supercapacitor NiMo-LDH@NiCo-MOF//AC was assembled by using NiMo-LDH@NiCo-MOF as the positive electrode and commercial, activated carbon as the negative electrode [34,35]. Appendix A shows the CV images of different samples in two- and three-electrode systems for comparison. The figure presents the CV curves of NiMo-LDH, NiCo-MOF, and NiMo-LDH@NiCo-MOF electrode materials at the same scanning rate; NiMo-LDH@NiCo-MOF has a larger area than the other two electrodes. The results show that NiMo-LDH@NiCo-MOF has more electrochemically active sites, a higher specific capacity, and less polarization, indicating that the synergistic effect of Ni, Co, and Mo elements plays a key role in reducing the polarization. As shown in Figure 10a, at the scanning rate of 20 mV s^−1^, NiMo-LDH@NiCo-MOF and activated carbon electrode materials have potential windows of 0–0.5 and −1–0 V, respectively. Additionally, the operating voltage of NiMo-LDH@NiCo-MOF and AC is complementary, so the operating voltage of ASC can reach 1.6 V. Appendix A shows the CV and GCD tests under different potential windows in the dual-electrode test. The CV curves of ASC devices tested in different voltage windows show that the NiMo-LDH@NiCo-MOF//AC stable operating voltage window can reach 1.6 V, and a series oF·gCD curves show further proof of this point. Therefore, the actual operating voltage window of ASC is set to 0–1.6 V. When the scanning rate ranges from 5 to 100 mV·s^−1^, the CV curve of ASC is shown in Figure 10b. The quasi-rectangular geometry of the CV curve indicates that the ASC device has double-layer capacitance and pseudo-capacitance behavior. In addition, Figure 10c shows the variation in the charge and discharge curves of the ASC device at different current densities under a voltage window of 1.6 V. It can be observed that the GCD curves have symmetry, which confirms the excellent electrochemical reversibility and high coulomb efficiency of the ASC device [36]. Figure 10d shows that the specific capacitances of 167, 142, 117, 99, 87, and 74 F·g^−1^ are obtained for the GCD curves of ASC when the current densities are 1, 2, 4, 6, 8, and 10 A·g^−1^, respectively. The Ragone diagram of ASC devices is an important factor to evaluate the energy storage performance [37]. As can be observed from Figure 10e, a high-energy density value of 73 Wh·kg^−1^ is obtained at a power density of 802 W·kg^−1^, while a value of 39 Wh·kg^−1^ is retained at a high power density of nearly 8 kW·kg^−1^. In addition, NiMo-LDH@NiCo-MOF//AC ASC outperforms many other asymmetric supercapacitors using MOF-based electrode materials, such as NiCo-MOF/NF//AC (34.3 Wh·kg^−1^ at 375 W·kg^−1^) [30], NiCo-LDH//AC (27.5 Wh·kg^−1^ at 375 W·kg^−1^) [38], NiCo-MOF//AC (20.94 Wh·kg^−1^ at 750.84 W·kg^−1^) [8], and Ni/Co-MOF//AC (56 Wh·kg^−1^ at 800 W·kg^−1^) [10]. Figure 10f shows the cycle life of the charge and discharge of the device at 10 A·g^−1^. From 2000 cycles onwards, the specific capacitance is improved, which may be because the active site inside the NiMo-LDH@NiCo-MOF//AC electrode is fully contacted in the electrolyte, making the redox reaction more thorough and confirming the excellent cycling stability. For NiMo-LDH@NiCo-MOF//AC, an initial capacity retention rate of 87.6% was achieved after 10,000 cycles, indicating its potential use in practical applications. At the same time, the coulomb efficiency of NiMo-LDH@NiCo-MOF//AC during the whole cycle is close to 100%, showing good stability and a reversible redox reaction [39]. The long-term cycling stability of the HSC device is due to the optimized electrode structure and large surface area, and promotes the interface contact efficiency of the electrode and electrolyte, and provides sufficient volume expansion space during the charge–discharge cycle.

## 4. Conclusions

NiMo-LDH@NiCo-MOF was successfully synthesized with 3D interconnected lamellar structures using a simple hydrothermal method, in which NiCo-MOF was grown and directionally immobilized on LDHs. The flower-like NiMo-LDH structure serves as a skeleton and conduction pathway, providing efficient charge transport for Farrah reactions and strong support for structural integrity. In addition, the space between the sheets effectively mitigated the volume variation during OH^−^ insertion and extraction processes. The transition metals Ni, Co, and Mo combined to provide more active sites, thereby improving electrical conductivity. Therefore, the synergistic effect of NiMo-LDH and NiCo-MOF can ensure a large interfacial contact area and short charge transport distance of the composite nanosheets, resulting in the excellent electrochemical behavior of NiMo-LDH@NiCo-MOF. The prepared NiMo-LDH@NiCo-MOF electrode not only provides enhanced conductivity and cyclic stability, but also achieves excellent specific capacitance, achieving an excellent performance of 1536 F·g^−1^ at 1 A·g^−1^. During charge storage, the nanoflower structure of NiMo-LDH@NiCo-MOF can provide more active sites for electrochemical reactions, further expand the ion transport channel, and show better electrochemical kinetics and pseudo-capacitance performance. In addition, NiMo-LDH@NiCo-MOF asymmetrical supercapacitors can achieve an excellent energy density of 60.2 Wh·kg^−1^ at 797 W·kg^−1^ and have satisfactory cycle stability after 10,000 cycles, with a capacitance retention rate of 87.6%. It is shown that the NiMo-LDH@NiCo-MOF electrode presents a broad prospect for its use as a high-performance supercapacitor.

## Figures and Tables

**Figure 1 nanomaterials-13-00581-f001:**
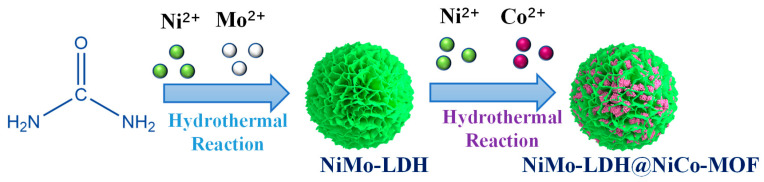
Schematic diagram of the synthesis method for NiMo-LDH@NiCo-MOF.

**Figure 2 nanomaterials-13-00581-f002:**
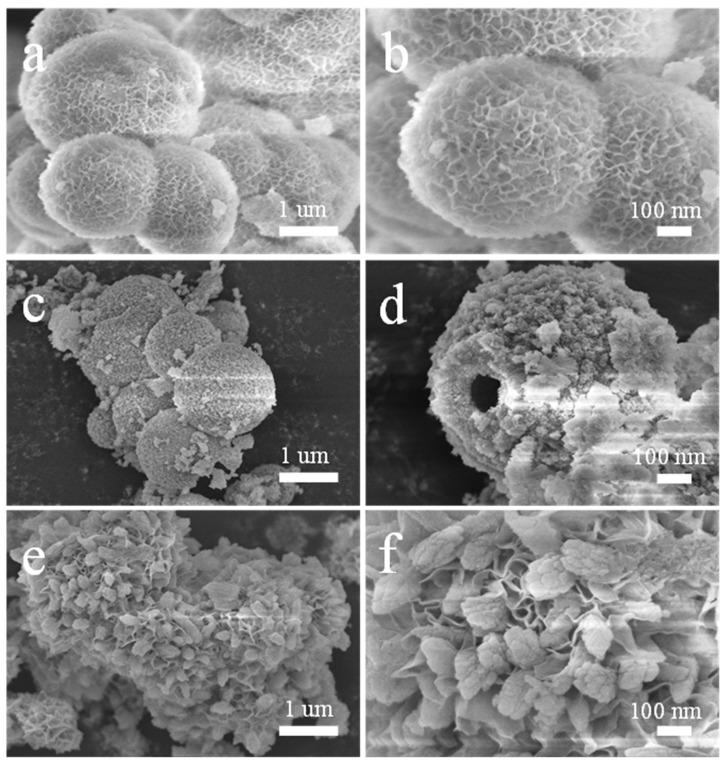
SEM images of the prepared NiMo-LDH (**a**,**b**), NiCo-MOF(**c**,**d**), and NiMo-LDH@NiCo-MOFs (**e**,**f**) under different magnifications.

**Figure 3 nanomaterials-13-00581-f003:**
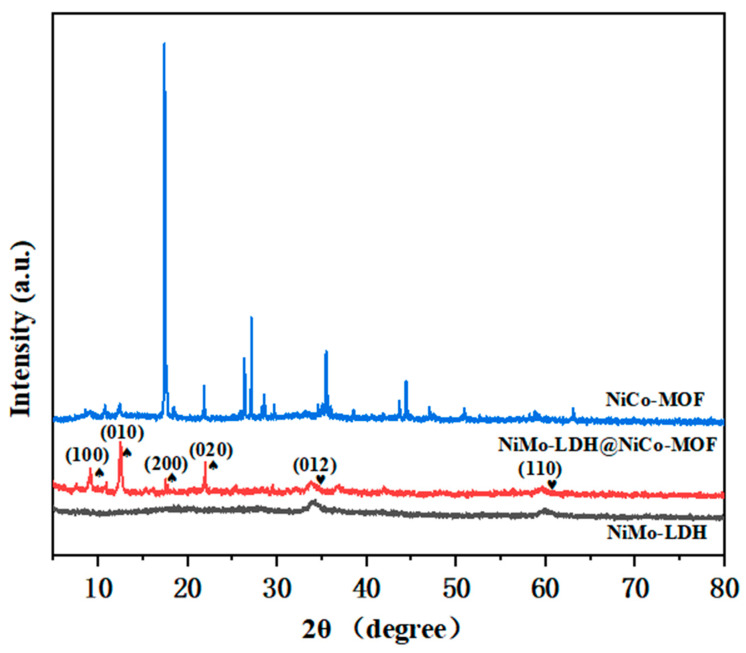
X-ray diffraction patterns of different samples.

**Figure 4 nanomaterials-13-00581-f004:**
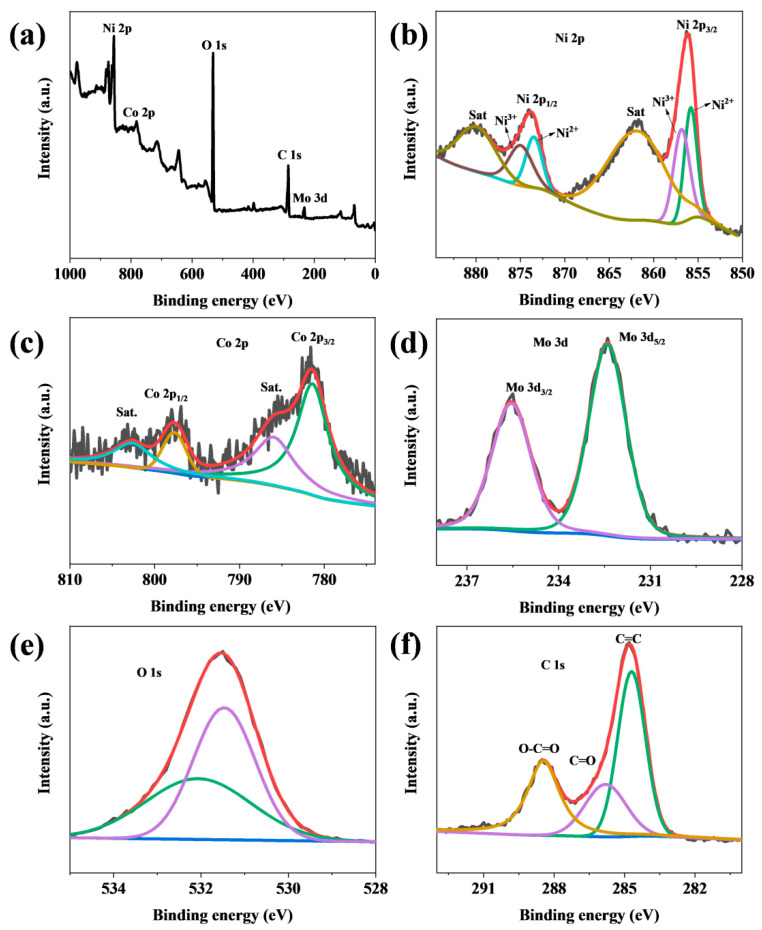
(**a**) XPS survey of the NiMo-LDH@NiCo-MOF, (**b**) Ni 2p, (**c**) Co 2p, (**d**) Mo 3d, (**e**) O 1s, and (**f**) C 1s. In the Ni 2p spectrum, Ni^2+^ corresponds to green and blue, and Ni^3+^ corresponds to purple and dark brown. Satellite peaks correspond to light brown, and red corresponds to fitted peaks. In the Co 2p spectrum, green corresponds to Co 2p3/2, brown to Co 2p1/2, purple and blue to the satellite peaks of Co 2p, and red to the fitted peaks. In the spectrum of Mo 3d, green corresponds to Mo 3d5/2, purple corresponds to Mo 3d3/2, and red corresponds to the fitted peak. In the spectrum of O 1s, purple corresponds to metal-oxygen bonds in metal oxides, and green corresponds to oxygen in hydroxyl ions. The red corresponds to the peaks of the fit. In the spectrum of C 1s, green corresponds to the C=C bond, purple corresponds to C=O, brown corresponds to the O-C=O bond, and red corresponds to the fitted peak.

**Figure 5 nanomaterials-13-00581-f005:**
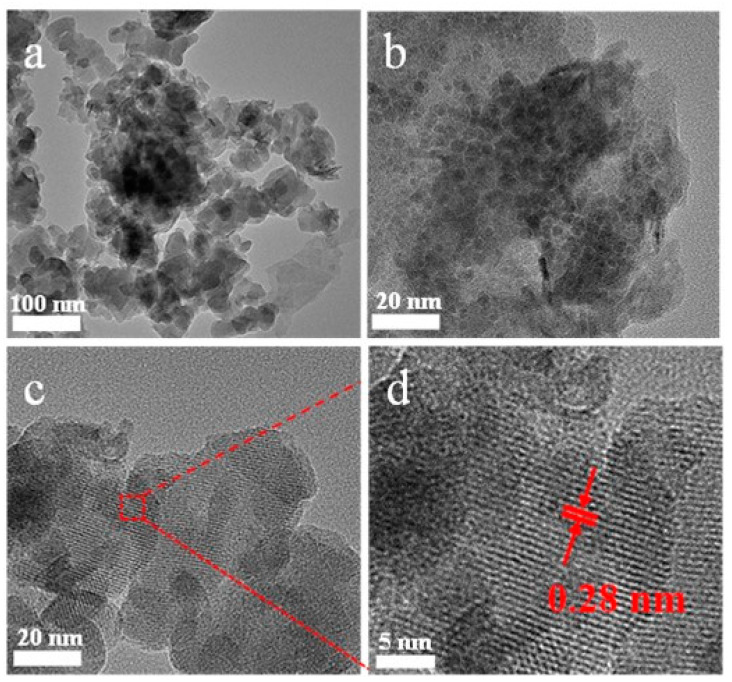
(**a**–**d**) TEM and HRTEM images of the prepared NiMo-LDH@NiCo-MOF.

**Figure 6 nanomaterials-13-00581-f006:**
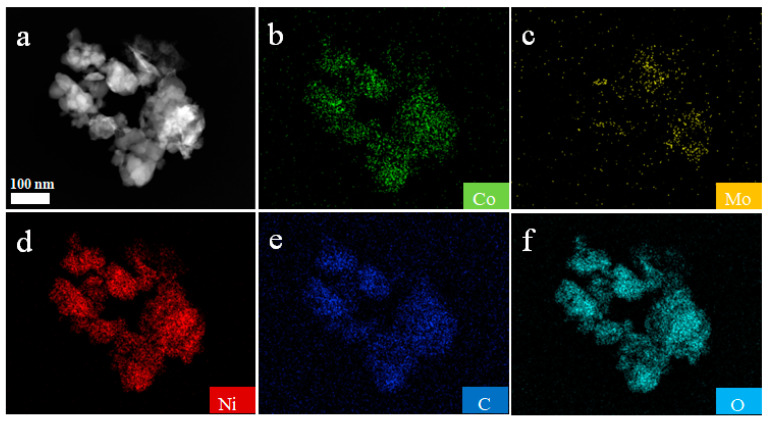
The elementary mapping of NiMo-LDH@NiCo-MOF, (**a**–**f**) shows that the product is composed of Co, Ni, Mo, C, O, and other elements, and each element is evenly distributed.

**Figure 7 nanomaterials-13-00581-f007:**
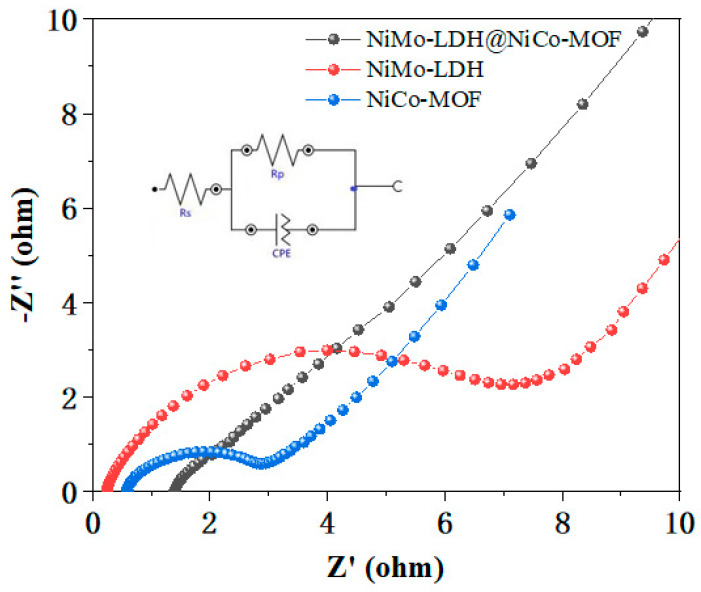
Nyquist plots of NiCo-MOF, NiMo-LDH, NiMo-LDH@NiCo-MOF.

**Figure 8 nanomaterials-13-00581-f008:**
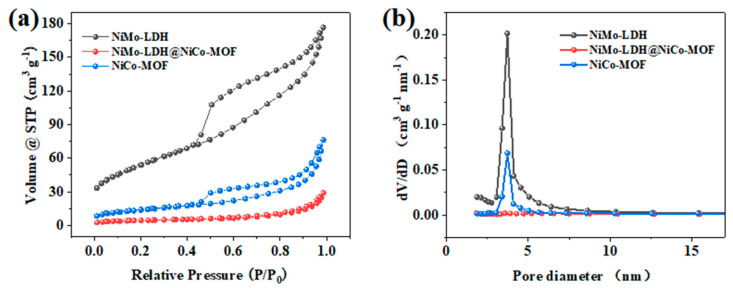
(**a**) Nitrogen adsorption and desorption curves; (**b**) pore size distribution.

**Figure 9 nanomaterials-13-00581-f009:**
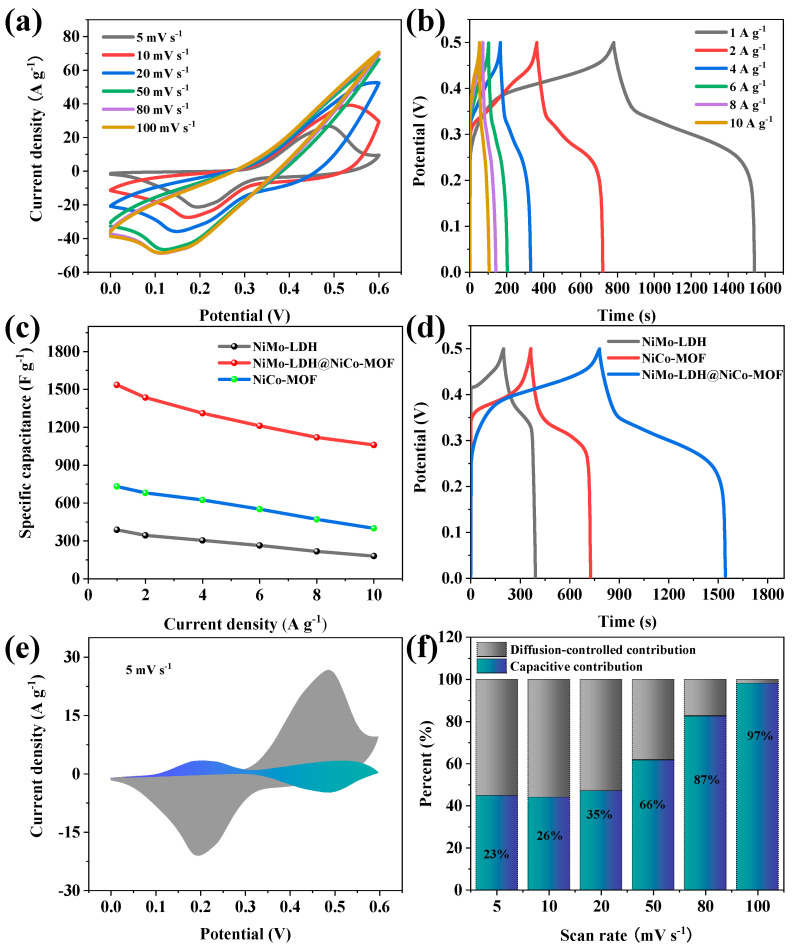
(**a**) CV curves of NiMo-LDH@NiCo-MOF; (**b**) GCD curves of NiMo-LDH@NiCo-MOF at 1-10 A·g^−1^; (**c**) specific capacitance of NiMo-LDH@NiCo-MOF, NiMo-LDH, and NiCo-MOF; (**d**) GCD curves; (**e**) capacitance and diffusion contribution at 5 mV s^−1^; (**f**) proportion of capacitance contribution.

**Figure 10 nanomaterials-13-00581-f010:**
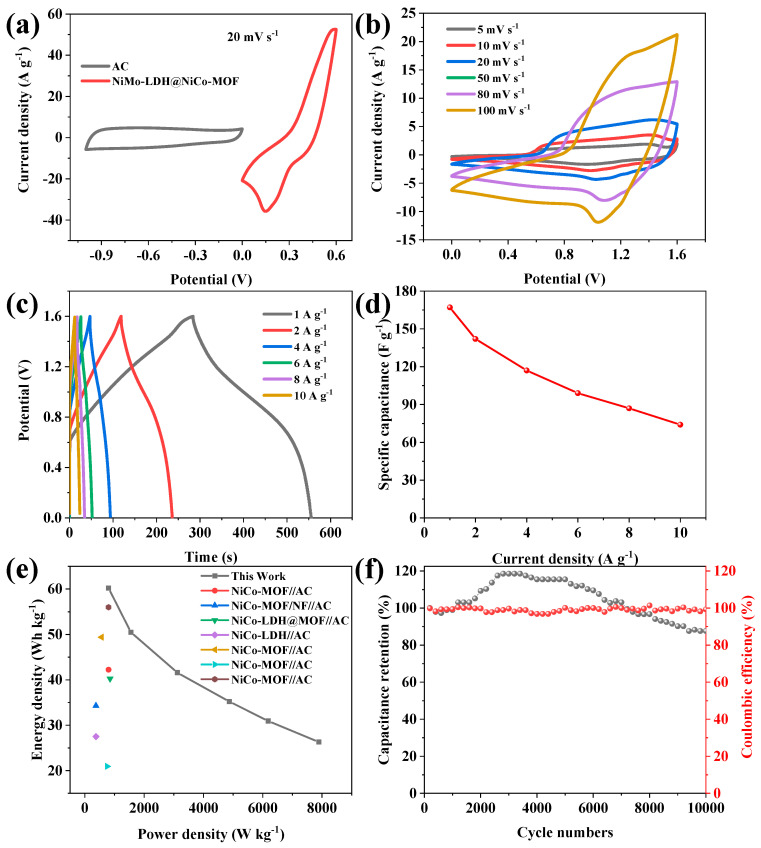
(**a**) CV curves of NiMo-LDH@NiCo-MOF and AC electrodes; (**b**) CV curve of NiMo-LDH@NiCo-MOF//AC; (**c**) GCD curves of NiMo-LDH@NiCo-MOF//AC; (**d**) specific capacitance of NiMo-LDH@NiCo-MOF//AC; (**e**) Ragone plot; (**f**) cyclic stability and coulomb effect diagram at 10 A·g^−1^.

## Data Availability

Data will be made available on request.

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
