# Peer review of "In Situ Growth of Nickel–Cobalt Metal Organic Frameworks Guided by a Nickel–Molybdenum Layered Double Hydroxide with Two-Dimensional Nanosheets Forming Flower-Like Struc-Tures for High-Performance Supercapacitors"

_nanomaterials, 2023, doi:10.3390/nano13030581_

Round 1

Reviewer 1 Report

Review Comments

Cheng Cheng et. al report title “nickel-cobalt metal organic frameworks guided by nickel-molybdenum layered hydroxide with two-di-mensional nanosheets forming flower-like structures for high-performance supercapacitors”, This is an interesting study and the authors have  The paper is generally well written and structured. However, in my opinion, the paper has after minor revision considered for publication.

1.    Abstract: Please focus the abstract on MOF nanoparticles and LDH nanosheets. In particular, what is the LDH abbreviation?

2.    What is the final product yield of NiMo-LDH@NiCo-MOF composites?

3.    Page No.4, Fig. 3 XRD, please confirm the reference ICC or CCDC, for example; “which is consistent with the NiCo-MOF card (CCDC 638866) indexed [8]”

4.    10 (f) cyclic stability and coulomb effect not acceptable, Author’s should be revised cycling stability for capacitance retention, exceeding more than 100%, figure 10 f, not acceptable.

5.    Reviewer appreciates very good achieving an excellent energy density ED - 60.2 Wh/kg.

Author Response

  1. Abstract: Please focus the abstract on MOF nanoparticles and LDH nanosheets. In particular, what is the LDH abbreviation?

Response: Thank you four your suggestion. LDH stands for layered double hydroxide.

  1. What is the final product yield of NiMo-LDH@NiCo-MOF composites?

Response: Thank you for your question. The final product yield of NiMo-LDH@NiCo-MOF composites is about 0.2g.

  1. Page No.4, Fig. 3 XRD, please confirm the reference ICC or CCDC, for example; “which is consistent with the NiCo-MOF card (CCDC 638866) indexed [8]”

Response: Thank you for your comments. ICC cards correspond to NiMo-LDH and CCDC to NiCo-MOF. It shows that the two substances are successfully synthesized together.

  1. 10 (f) cyclic stability and coulomb effect not acceptable, Author’s should be revised cycling stability for capacitance retention, exceeding more than 100%, figure 10 f, not acceptable.

Response: Thank you for your comments. After 2000 cycles, the electrode material was fully activated, so the capacity retention rate increased. The coulomb efficiency wave exceeds 100% because the electrode material contains nitrogen and oxygen elements at high current density, and some side reactions occur in the electrochemical redox reaction, leading to incomplete discharge. As the cycle progresses, the charging and discharging time is shortened and the coulomb efficiency increases.

  1. Reviewer appreciates very good achieving an excellent energy density ED - 60.2 Wh/kg.

Response: Thank you for your appreciation. The three-dimensional interconnected sheet structure of NiMo-LDH@NiCo-MOF ensures a large interfacial contact area and short charge transfer distances for the composite nanosheets, resulting in an excellent energy density of NiMo-LDH@NiCo-MOF.

Reviewer 2 Report

This paper reports the fabrication of NiMo-LDH@NiCo-MOF composite by a two-step method for supercapacitor application. The prepared NiMo-LDH@NiCo-MOF composite has been characterized by the authors using various techniques, including TEM, EDS, XPS, XRD and SEM. This composite was shown to exhibit higher capacitive performance than pure NiCo-MOF and NiMo-LDH. Furthermore, the asymmetric supercapacitor fabricated using the NiMo-LDH@NiCo-MOF and activated carbon showed a maximum energy density of 60.2 Wh/kg at a power density of 797 W/kg. This work shows the benefit of hybridizing bimetallic LDH material with bimetallic MOF for enhancing the electrochemical performance for supercapacitors. Overall, this work is suitable for publication in this journal after addressing the following comments:

1. In the Experimental section, the purity of the chemicals used need to be specified.

2. In the Experimental section, the synthetic procedures for preparing the pure NiCo-MOF should also be described.

3. The XPS of the NiMo-LDH@NiCo-MOF should be compared to the pure NiMo-LDH to see if any shift in the metal species peak is observed.

4. The morphology of the NiMo-LDH@NiCo-MOF after cycling should be check and presented to see the structural stability of this composite.

5. In Fig. 5f, the right side y-axis is missing the axis label. Please remedy this issue.

6. In the Introduction, some recent and relevant references on the fabrication of LDH-based electrode materials for supercapacitors, such as J. Colloid Interface Sci., 609, 114-129 (2022); ChemSusChem 13 (6), 1645-1655 (2020) and Nano Energy, 91, 106633 (2022) can be included and cited.

7. The assignment of the lattice spacing in the HRTEM image should be double checked. It is is better to find lattice spacing corresponding to the lattice plane of NiMo-LDH or NiCo-MOF rather than Co-MOF.

Author Response

  1. In the Experimental section, the purity of the chemicals used need to be specified.

Response: Thanks for your careful review. We have stated the purity of the chemical in the article.

  1. In the Experimental section, the synthetic procedures for preparing the pure NiCo-MOF should also be described.

Response: Thank you for your comments. We have added instructions in the experimental section.

  1. The XPS of the NiMo-LDH@NiCo-MOF should be compared to the pure NiMo-LDH to see if any shift in the metal species peak is observed.

Response: Thank you for your suggestion. The XPS spectra of NiMo-LDH are provided in the supporting information. The peak shapes of Ni 2p and Mo 3d do not change significantly compared with NiMo-LDH@NiCo-MOF, and the binding energies do not exceed 0.5eV.

  1. The morphology of the NiMo-LDH@NiCo-MOF after cycling should be check and presented to see the structural stability of this composite.

Response: Thank you for your suggestion. We have provided the morphology of the cyclic NiMo-LDH@NiCo-MOF in the supporting information. Although the structure of NiMo-LDH@NiCo-MOF after circulation has a little caking and collapse, the overall structural integrity is still maintained.

  1. In Fig. 5f, the right side y-axis is missing the axis label. Please remedy this issue.

Response: Thank you for your careful review. We have added the Y-axis label in Figure 10f.

  1. In the Introduction, some recent and relevant references on the fabrication of LDH-based electrode materials for supercapacitors, such as J. Colloid Interface Sci., 609, 114-129 (2022); ChemSusChem 13 (6), 1645-1655 (2020) and Nano Energy, 91, 106633 (2022) can be included and cited.

Response: Thank you for your suggestion. These references are valuable. We have added highly related papers to enrich the introduction according your suggestion.

  1. The assignment of the lattice spacing in the HRTEM image should be double checked. It is is better to find lattice spacing corresponding to the lattice plane of NiMo-LDH or NiCo-MOF rather than Co-MOF.

Response: Thank you for your suggestion. In previous literature, no corresponding lattice spacing was found, so Co-MOF was chosen as an alternative, but the successful synthesis of NiMo-LDH@NiCo-MOF can be proved by xrd crystal faces.